# Bis(chloroacetamidino)-Derived Heteroarene-Fused Anthraquinones Bind to and Cause Proteasomal Degradation of tNOX, Leading to c-Flip Downregulation and Apoptosis in Oral Cancer Cells

**DOI:** 10.3390/cancers14194719

**Published:** 2022-09-28

**Authors:** Jeng Shiun Chang, Chien-Yu Chen, Alexander S. Tikhomirov, Atikul Islam, Ru-Hao Liang, Chia-Wei Weng, Wei-Hou Wu, Andrey E. Shchekotikhin, Pin Ju Chueh

**Affiliations:** 1Department of Otolaryngology, Head and Neck Surgery, Jen-Ai Hospital, Taichung 41265, Taiwan; 2Institute of Biomedical Sciences, National Chung Hsing University, 145 Xingda Rd., Taichung 40227, Taiwan; 3Gause Institute of New Antibiotics, 11 B. Pirogovskaya Street, 119021 Moscow, Russia; 4Institute of Medicine, Chung Shan Medical University, Taichung 40201, Taiwan; 5Department of Post-Baccalaureate Medicine, College of Medicine, National Chung Hsing University, 145 Xingda Rd., Taichung 40227, Taiwan; 6Department of Medical Research, China Medical University Hospital, Taichung 40402, Taiwan; 7Graduate Institute of Basic Medicine, China Medical University, Taichung 40402, Taiwan

**Keywords:** anthraquinone, apoptosis, c-Flip downregulation, heterocyclic compounds, oral cancer cells, tumor-associated NADH oxidase (tNOX, ENOX2)

## Abstract

**Simple Summary:**

New-generation anthraquinone derivatives attached with different heterocycles and bearing chloroacetamidines in the side chains have been synthesized to reduce side effects and drug resistance. In this study, we identified the cellular target of the studied compounds through ligand binding assays and in silico simulations. Our results illustrate that the studied compounds bound to and targeted the tumor-associated NADH oxidase (tNOX) in oral cancer cells. tNOX is a growth-related protein and is found to be expressed in cancer cells but not in non-transformed cells, and its knockdown by RNA interference in tumor cells overturns cancer phenotypes, supporting its role in cellular growth. We also identified that tNOX bound to the studied compounds and underwent degradation, which was correlated with apoptosis induction in oral cancer cells.

**Abstract:**

Anthraquinone-based intercalating compounds, namely doxorubicin and mitoxantrone, have been used clinically based on their capacity to bind DNA and induce DNA damage. However, their applications have been limited by side effects and drug resistance. New-generation anthraquinone derivatives fused with different heterocycles have been chemically synthesized and screened for higher anticancer potency. Among the compounds reported in our previous study, 4,11-bis(2-(2-chloroacetamidine)ethylamino)anthra[2,3-b]thiophene-5,10-dione dihydrochloride (designated **2c**) was found to be apoptotic, but the direct cellular target responsible for the cytotoxicity remained unknown. Here, we report the synthesis and anticancer properties of two other derivatives, 4,11-bis(2-(2-chloroacetamidine)ethylamino)naphtho[2,3-f]indole-5,10-dione dihydrochloride (**2a**) and 4,11-bis(2-(2-chloroacetamidine)ethylamino)-2-methylanthra[2,3-b]furan-5,10-dione dihydrochloride (**2b**). We sought to identify and validate the protein target(s) of these derivatives in oral cancer cells, using molecular docking simulations and cellular thermal shift assays (CETSA). Our CETSA results illustrate that these derivatives targeted the tumor-associated NADH oxidase (tNOX, ENOX2), and their direct binding downregulated tNOX in p53-functional SAS and p53-mutated HSC-3 cells. Interestingly, the compounds targeted and downregulated tNOX to reduce SIRT1 deacetylase activity and increase Ku70 acetylation, which triggers c-Flip ubiquitination and induces apoptosis in oral cancer cells. Together, our data highlight the potential value of these heteroarene-fused anthraquinones in managing cancer by targeting tNOX and augmenting apoptosis.

## 1. Introduction

Anthraquinone (anthracene-9,10-dione)-based antitumor compounds have been utilized clinically for decades due to their high effectiveness against solid and hematological malignancies. The capacity of some representatives, including doxorubicin and mitoxantrone, to bind DNA and induce DNA damage/strand breaks is applied for tumor growth suppression [1,2,3]. However, the therapeutic action of these drugs is accompanied by side effects and the emergence of drug resistance. To address these limitations, researchers have sought to design and synthesize new generations of anthraquinone derivatives and screen them for improved anticancer properties [4,5]. Biological investigations revealed that these compounds target various molecules that are essential for tumor progression, including different secondary structures of nucleic acids, DNA topoisomerases, HDAC-6, and others [6,7,8,9,10,11,12]. Among the new-generation compounds, heterocyclic derivatives of anthraquinone are considered to be one of the most promising classes for the analog-based drug design of anticancer compounds [13,14,15]. In particular, the annellation of five-membered heterocycles with one heteroatom to 1,4-diaminoanthraquinone (an analog of mitoxantrone) yielded derivatives that retained a powerful antiproliferative profile and exhibited significantly greater activity against MDR-positive tumor cells [16]. In-depth studies showed that these compounds exerted multitarget anticancer action against tumor cells, simultaneously interacting with duplexes and G-quadruplexes of DNA, topoisomerase 1 (Top1), and Aurora B kinase to offer a potential new approach to cancer management [17,18,19,20].

Previously, we reported a set of 4,11-diaminoanthra[2,3-b]furan-5,10-diones and revealed a link between their antiproliferative potency and their ability to modulate the tNOX–NAD^+^–SIRT1 regulatory axis [21,22]. tNOX is a growth-related tumor-associated NADH (or hydroquinone) oxidase (ENOX2) that catalyzes the generation of NAD^+^ and is expressed in cancer cells but not in non-transformed cells [23,24,25,26]. tNOX deficiency caused by RNA interference in tumor cells can overturn cancer phenotypes, decreasing NAD^+^ generation, proliferation, and migration/invasion while increasing apoptosis; thus, tNOX confers a growth advantage [27,28,29,30,31,32]. Given that the acetylation levels of various proteins impact many cellular functions, we further explored the functional significance of tNOX-modulated NAD^+^ and NAD^+^-dependent deacetylase, SIRT1, in cancer cells [21,28,32,33,34]. In parallel, the clinical relevance of tNOX protein expression was corroborated in serum samples from individuals with a diagnosis of malignant mesothelioma, suggesting its role as a potential malignancy detection marker [35]. By data mining in the Kaplan–Meier plotter (pan-cancer RNA-seq dataset; www.kmplot.com (accessed on 13 March 2019)), we also observed that a high tNOX expression level was correlated with a poor prognosis for overall survival in male patients with liver cancer [22]. Excitingly, numerous anticancer agents, including oxaliplatin and doxorubicin, were reported to abolish tNOX activity or downregulate tNOX expression, and this was almost invariably associated with cancer cell apoptosis and/or autophagy [21,22,29,33,34,36,37,38,39].

We observed that a substitution in the terminal group of the 4,11-side chains in generated anthraquinones seemed to play a meaningful role in tNOX/SIRT1 inhibition and apoptosis induction; in particular, the introduction of methyl or 2-hydroxy radicals in the terminal amines contributed to tNOX downregulation, whereas guanidination dramatically abrogated this property [21,22]. Taking advantage of the ability of highly basic chloroacetamidine groups to form several H-bonds and strong ionic interactions to improve binding affinity and highly lipophilic acetamidines to penetrate into tumor cells [40], we herein synthesized and evaluated the anticancer activity of anthraquinone derivatives fused with different heterocycles and bearing 2-chloroacetamidine groups in the terminal positions of their side chains. We found that tNOX bound with these derivatives and underwent downregulation, and this was correlated with apoptosis induction in p53-functional SAS and p53-mutated HSC-3 oral cancer cells.

## 2. Materials and Methods

### 2.1. Chemistry

All solvents, chemicals, and reagents were obtained commercially and used without purification. Analytical TLC was performed on Silica Gel F_254_ plates (Merck), and column chromatography was performed using SilicaGel Merck 60. Melting points were determined using a Buchi SMP-20 apparatus and are given uncorrected. ^1^H and ^13^C NMR spectra were recorded on a Varian Mercury 400 Plus instrument operated at 400 MHz and 100 MHz, respectively. Chemical shifts were measured in DMSO-*d*_6_ using tetramethylsilane as an internal standard. High-resolution mass spectra were recorded with electron spray ionization (ESI) on a Bruker Daltonics microOTOF-QII instrument. HPLC was performed using a Shimadzu Class-VP V6.12SP1 system. All solutions were evaporated under reduced pressure using an IKA RV 10 rotary evaporator at <45 °C. All products were vacuum-dried at room temperature. The purity of the final samples of 4,11-bis(2-(2-chloroacetamidine)ethylamino)naphtho[2,3-f]indole-5,10-dione dihydrochloride (designated derivative **2a**), 4,11-bis(2-(2-chloroacetamidine)ethylamino)-2-methylanthra[2,3-b]furan-5,10-dione dihydrochloride (derivative **2b**), and 4,11-bis(2-(2-chloroacetamidine)ethylamino)anthra[2,3-b]thiophene-5,10-dione dihydrochloride (derivative **2c**) was >95%, as determined by HPLC analysis.

### 2.2. 4,11-bis(2-(2-chloroacetamidine)ethylamino)naphtho[2,3-f]indole-5,10-dione dihydrochloride (***2a***)

A solution of 4,11-bis[(2-aminoethyl)amino]naphtho[2,3-f]indole-5,10-dione (**1a**, 100 mg, 0.26 mmol) [41], ethyl 2-chloroacetimidate hydrochloride (0.2 g, 1.3 mmol), and diisopropylethylamine (DIPEA, 0.23 mL, 1.3 mmol) in a mixture of DMSO (10.0 mL) and methanol (10.0 mL) was stirred for 3 h at 50 °C. Reaction mixture was cooled and quenched with acetone. The dark blue precipitate was filtered off and purified by column chromatography on silica gel (CHCl_3_–MeOH–HCO_2_H, 80:20:2 *v*/*v*). The residue was dissolved in a mixture of water (1 mL) and MeOH (3 mL), and a solution of 1 N HCl in MeOH (0.2 mL) was added. The product was precipitated with an excess of acetone, filtered off, washed with acetone (10 mL) and ether (5 mL), and dried in vacuum. The yield of compound **2a** as dihydrochloride salt was 71 mg (47%) as a dark blue powder, mp > 250 °C (decomp). HPLC Kromasil-100-5-mkm C-18 column (4.6 × 250 mm, LW = 260 nm), eluent: A—H_3_PO_4_ (0.01 M), B—MeCN; gradient B 15→40% (20 min), 40→70% (10 min), elution time 12.1 min, purity 98%. ^1^H NMR (400 MHz, DMSO-*d_6_*) δ 12.45 (s, 1H, 1-H); 10.42 (s, 1H, Ar-NH); 10.31 (s, 1H, Ar-NH); 9.82 (s, 1H, NH); 9.77 (s, 1H, NH); 9.73 (s, 1H, NH); 9.66 (s, 1H, NH); 8.28–8.24 (m, 2H, 6,9-H); 7.75–7.71 (m, 2H, 7,8-H); 7.56 (s, 1H, 3-H); 7.14 (s, 1H, 2-H); 4.42 (2H, s, CH_2_Cl); 4.30 (2H, s, CH_2_Cl); 4.11 (4H, br s, 2CH_2_); 3.81–3.72 (4H, m, 2CH_2_). ^13^C NMR (100 MHz, DMSO-*d*_6_) δ 179.4 (C=O), 177.8 (C=O), 162.5 (C), 162.3 (C), 147.8 (C), 141.9 (C), 134.8 (C), 134.3 (C), 131.8 (CH), 131.4 (CH), 128.2 (CH), 128.1 (CH), 125.7 (2CH), 107.6 (C), 106.5 (C), 103.4 (C), 46.5 (CH_2_), 45.4 (CH_2_), 43.6 (CH_2_), 43.2 (CH_2_), 42.3 (CH_2_), 42.2 (CH_2_). HRMS (ESI) calculated for C_24_H_25_Cl_2_N_6_O_2_S C_24_H_26_Cl_2_N_7_O_2_ [M + H]^+^ 514.1520, found 514.1535.

### 2.3. 4,11-bis(2-(2-chloroacetamidine)ethylamino)-2-methylanthra[2,3-b]furan-5,10-dione dihydrochloride (***2b***)

This compound was prepared from **1b** [21] as described for **2b**. The yield of compound **2b** as dihydrochloride salt was 78 mg (50%) as a dark blue powder, mp > 250 °C (decomp). HPLC Kromasil-100-5-mkm C-18 column (4.6 × 250 mm, LW = 260 nm), eluent: A—H_3_PO_4_ (0.01 M), B—MeCN; gradient B 15→40% (20 min), 40→70% (10 min), elution time 13.3 min, purity 95%. ^1^H NMR (400 MHz, CD_3_OD) δ 8.18–8.15 (m, 2H, 6,9-H); 7.69–7.67 (m, 2H, 7,8-H); 6.85 (s, 1H, 3-H); 4.49 (s, 2H, CH_2_Cl); 4.41 (s, 2H, CH_2_Cl); 4.14–4.11 (2H, m, CH_2_); 3.99–3.97 (2H, m, CH_2_); 3.79–3.77 (2H, m, CH_2_); 3.73–3.71 (2H, m, CH_2_); 2.47 (3H, s, CH_3_). ^1^H NMR (400 MHz, DMSO-*d_6_*) δ 12.31 (br s, 1H, Ar-NH); 11.79 (br s, 1H, Ar-NH); 10.54 (s, 1H, NH); 10.43 (s, 1H, NH); 9.72–9.71 (m, 2H, NH_2_); 9.44 (s, 1H, NH_2_); 9.36 (s, 1H, NH_2_); 8.25–8.22 (m, 2H, 6,9-H); 7.76–7.74 (m, 2H, 7,8-H); 7.22 (s, 1H, 3-H); 4.50 (2H, s, CH_2_Cl); 4.48 (2H, s, CH_2_Cl); 4.09 (2H, br s, CH_2_); 3.98–3.94 (2H, m, CH_2_); 3.59–3.56 (H, m, CH_2_); 3.55–3.51 (H, m, CH_2_); 2.49 (3H, s, CH_3_). ^13^C NMR (100 MHz, DMSO-*d*_6_) δ 180.9 (C=O), 179.3 (C=O), 162.4 (C), 162.2 (C), 150.0 (C), 148.5 (C); 144.7 (CH); 136.7 (C), 134.1 (C), 133.8 (C); 132.7 (CH), 132.4 (CH), 125.9 (CH), 125.8 (CH), 121.1 (C); 111.9 (C); 108.4 (C); 105.6 (C); 46.9 (CH_2_), 45.8 (CH_2_), 44.4 (CH_2_), 44.1 (CH_2_), 41.9 (CH_2_), 41.0 (CH_2_); 13.6 (CH_3_). HRMS (ESI) calculated for C_25_H_27_Cl_2_N_6_O_3_ [M + H]^+^ 529.1516, found 529.1542.

### 2.4. 4,11-bis(2-(2-chloroacetamidine)ethylamino)anthra[2,3-b]thiophene-5,10-dione dihydrochloride (***2c***)

This compound was prepared from **1c** [42] as described for **2c**. The yield of compound **2c** as dihydrochloride salt was 78 mg (45%) as a dark blue powder, mp > 250 °C (decomp). HPLC Kromasil-100-5-mkm C-18 column (4.6 × 250 mm, LW = 260 nm), eluent: A—H_3_PO_4_ (0.01 M), B—MeCN; gradient B 10→50% (30 min), elution time 15.9 min, purity 95%.^1^H NMR (400 MHz, DMSO-*d_6_*) δ 11.93 (br s, 2H, 2Ar-NH); 10.73–10.70 (m, 2H, 2NH); 10.00–9.98 (m, 2H, NH_2_); 9.67–9.65 (m, 2H, NH_2_); 8.19–8.16 (m, 3H, 2,6,9-H); 7.95 (d, 1H, J = 4.4, 3-H); 7.74–7.72 (m, 2H, 7,8-H); 4.54 (2H, s, CH_2_Cl); 4.53 (2H, s, CH_2_Cl); 4.10–4.09 (2H, m, CH_2_); 3.95–3.94 (2H, m, CH_2_); 3.73–3.70 (4H, m, 2CH_2_). ^13^C NMR (100 MHz, DMSO-*d*_6_) δ 180.7 (2C=O), 163.0 (C), 162.9 (C), 146.0 (C), 145.1 (C), 135.5 (C), 135.2 (C), 134.0 (C), 133.9 (C), 108.2 (C), 106.8 (C), 132.4 (2CH), 131.4 (CH), 126.0 (CH), 125.8 (2CH), 45.0 (CH_2_), 43.2 (CH_2_), 43.1 (CH_2_), 43.0 (CH_2_), 38.9 (2CH_2_). HRMS (ESI) calculated for C_24_H_25_Cl_2_N_6_O_2_S [M + H]^+^ 531.1131, found 531.1140.

### 2.5. Cell Culture and Reagents

The anti-PARP, anti-Bcl2, anti-c-Flip, anti-ADP-ribose, anti-γH2AX, and anti-caspase 8 antibodies were purchased from Cell Signaling Technology, Inc. (Beverly, MA, USA). The anti-β-actin antibody was from Millipore Corp. (Temecula, CA, USA). The anti-ubiquitin antibody was purchased from Santa Cruz Biotechnology, Inc. (Santa Cruz, CA, USA). The anti-Ku70 and anti-acetyl-Ku70 antibodies were from Arigo Biolaboratories (Hsinchu, Taiwan). The commercially available anti-ENOX2 (a.k.a. tNOX, COVA1) antibody from Proteintech (Rosemont, IL, USA) was used for immunoprecipitation. The antisera to tNOX used for immunoblotting were generated as described previously [43]. The anti-mouse and anti-rabbit IgG antibodies and other chemicals were purchased from Sigma Chemical Company (St. Louis, MO, USA) unless otherwise specified.

SAS (human squamous cell carcinoma of the tongue) and HSC-3 (human tongue squamous cell carcinoma) cells were grown in Dulbecco’s Modified Eagle Medium (DMEM), and were kindly provided by Dr. Yuen-Chun Li (Department of Biomedical Sciences, Chung Shan Medical University, Taiwan). Media were supplemented with 10% FBS, 100 units/mL penicillin, and 50 µg/mL streptomycin. Cells were maintained at 37 °C in a humidified atmosphere of 5% CO_2_ in air, and the media were replaced every 2–3 days. Cells were treated with different concentrations of individual derivatives (dissolved in water), as described in the text, or with the same volume of water (vehicle control).

### 2.6. Continuous Monitoring of Cell Growth by Cell Impedance Measurements

For continuous monitoring of changes in cell growth, cells (7.5 × 10^3^ cells/well) were seeded onto E-plates and incubated for 30 min at room temperature, and then the E-plates were placed onto the xCELLigence System (Roche, Mannheim, Germany). Cells were grown overnight and then exposed to different derivatives or water, and impedance was measured every hour, as previously described [32]. Cell impedance was defined by the cell index (CI) = (Z_i_ − Z_0_) [Ohm]/15[Ohm], where Z_0_ is background resistance and Z_i_ is the resistance at an individual time point. A normalized cell index was determined as the cell index at a certain time point (CI_ti_) divided by that at the normalization time point (CI_nml_time_).

### 2.7. Apoptosis Determination

Apoptosis was measured using an Annexin V-FITC Apoptosis Detection Kit (BD Pharmingen, San Jose, CA, USA). Cells cultured in 6 cm dishes were trypsinized and collected by centrifugation. The cell pellet was washed, resuspended in 1× binding buffer, and stained with annexin V-FITC (fluorescein isothiocyanate), as recommended by the manufacturer. Cells were also stained with propidium iodide (PI) to detect necrosis or late apoptosis. Early apoptotic cells are Annexin V-FITC-positive and PI-negative, whereas late apoptotic cells are Annexin V-FITC and PI double-positive, analyzed using a Beckman Coulter FC500 flow cytometer.

### 2.8. Reverse Transcriptase–Polymerase Chain Reaction (RT-PCR)

Total RNA from oral cancer cells was isolated using the TRIzol reagent (Gibco, Carlsbad, CA, USA). First-strand cDNA was synthesized from 1 μg of total RNA using Superscript II (Life Technologies, Rockville, MD, USA). The primers for tNOX (NCBI Reference Sequence: XM_034949457.1) were used for PCR amplifications: 5′-GCTGTGCTTCTAGGCTGTGT-3′ (sense) and 5′-TTATCAAGACGGTGCAAGTAGGA-3′ (antisense), while GADPH (NCBI Reference Sequence: NM_001411843.1) were 5′-GGAAGGCCATGCCAGTGAGC-3′ (sense) and 5′-TATCGTGGAAGGACTCATGA-3′ (antisense). The reaction conditions consisted of 30 cycles of 95 °C for 30 s, 55 °C for 30 s, and 72 °C for 1 min, followed by a final extension of 5 min at 72 °C. The obtained PCR products were resolved by 1.4% agarose gel electrophoresis and visualized by DNA View nucleic acid staining.

### 2.9. Identification of tNOX as a Cellular Target by Cellular Thermal Shift Assays (CETSA)

The ability of each derivative to target intracellular tNOX was established by CETSA. Briefly, cells (2 × 10^7^) were seeded in 10 cm cultured dishes, cultured for 24 h, pretreated with 10 μM MG132 for 1 h, washed with PBS, treated with trypsin, and collected. The samples were centrifuged at 12,000× *g* rpm for 3 min at room temperature, the pellets were gently resuspended with 1 mL of PBS, and the samples were centrifuged at 7500× *g* rpm for 3 min at room temperature. The pellets were resuspended with 1 mL of PBS containing 20 mM Tris-HCl, pH 7.4, 100 mM NaCl, 5 mM EDTA, 2 mM phenylmethylsulfonyl fluoride (PMSF), 10 ng/mL leupeptin, and 10 μg/mL aprotinin. The samples were transferred to Eppendorf tubes and subjected to three freeze–thaw cycles; for each cycle, the tubes were exposed to liquid nitrogen for 3 min, placed in a heating block at 37 °C for 3 min, and vortexed briefly. For the experimental sample set, each individual derivative was added to a final concentration of 20 μM; for the control sample set, the same volume of vesicle solvent was added. The samples were heated at 37 °C for 1 h and dispensed to 100 μL aliquots. Pairs consisting of one control aliquot and one experimental aliquot were heated at 40 °C, 43 °C, 46 °C, 49 °C, 52 °C, 55 °C, 58 °C, 61 °C, or 67 °C for 3 min. Insoluble proteins were separated by centrifugation at 12,000× *g* rpm for 30 min at 4 °C, and the soluble protein-containing supernatants were resolved by SDS-PAGE and subjected to Western blot analysis using antisera to tNOX [27,43]. β-actin was detected as a loading control.

### 2.10. Immunoblotting and Immunoprecipitation

Cell extracts were prepared in lysis buffer (20 mM Tris-HCl pH 7.4, 100 mM NaCl, 5 mM EDTA, 2 mM phenylmethylsulfonyl fluoride (PMSF), 10 ng/mL leupeptin, 10 μg/mL aprotinin). Volumes of extract containing equal amounts of proteins (40 µg) were applied to SDS-PAGE gels, and resolved proteins were transferred to PVDF membranes (Schleicher & Schuell, Keene, NH, USA). The membranes were blocked, washed, and probed with a primary antibody. After washing to remove unbound primary antibody, membranes were incubated with horseradish peroxidase-conjugated secondary antibody for 1 h. The blots were washed again and developed using enhanced chemiluminescence (ECL) reagents, according to the manufacturer’s protocol (Amersham Biosciences, Piscataway, NJ, USA).

For immunoprecipitation, protein extracts from 100 mm dishes were incubated with 20 μL of Protein G Agarose Beads (for rabbit antibodies) for 1 h at 4 °C in rotation for pre-clearing. Then, 30 µg of supernatants was collected to use as input (control). Ubiquitin antibodies or control IgG were added into the supernatants and incubated onto beads in 500 μL of PBS-Tween 20 1% for overnight in rotation at 4 °C. Beads were precipitated by centrifugation at 3000× *g* rpm, 2 min at 4 °C. Beads were washed twice with lysis buffer and samples were prepared for Western blotting analysis.

### 2.11. Molecular Docking Simulation

The chemical structures of the derivatives were constructed using the MolView server and saved in SDF format [44]. The predicted full-length structure of human tNOX was downloaded from the AlphaFold Protein Structure Database and used for this docking study [45]. Molecular docking was performed with parameters of population size = 1500, generations = 80, and numbers of solutions = 3, and the docking energy was calculated using the iGEMDOCK software [46]. The docking poses of derivatives were analyzed using PyMol version 2.0.4 (https://pymol.org/2/ (accessed on 18 May 2022); PyMOL Molecular Graphics System, Schrodinger, New York, NY). The interaction diagram was generated using the BIOVIA Discovery Studio Visualizer (https://discover.3ds.com/discovery-studio-visualizer-download (accessed on 18 May 2022); Dassault System, San Diego, CA, USA).

### 2.12. Statistics

All data are expressed as the means ± SEs of three independent experiments. The significance of differences between control and treatment groups was calculated using a one-way ANOVA.

## 3. Results

### 3.1. Synthesis of Heteroarene-Fused Anthraquinones

Starting anthraquinone derivatives **1a**–**c**, which were annealed with pyrrole, furan, and thiophene, respectively, were synthesized in accordance with our previously reported multistep procedures [21,41,42]. The terminal amino groups of the 4,11-side chains can be converted to acetamidines by reaction with an appropriate iminoester [40,47]. Here, mild heating of amines **1a**–**c** with 2-chloroacetimidate hydrochloride in the presence of Hünig’s base (diisopropylethylamine, DIPEA) was used to generate 4,11-bis(2-(2-chloroacetamidine)ethyl)amine derivatives of heteroarene-fused anthraquinones (**2a**–**c**) at reasonable yields (Figure 1) (Figure 1). Column chromatography followed by treatment with 1N HCl in methanol was used to obtain water-soluble hydrochloride salts of **2a**–**c**. The assigned structures of **2a**–**c** were confirmed by ^1^H and ^13^C NMR; the purity of the final samples was >95% (Appendix A).

### 3.2. Bis(chloroacetamidino)heteroarene-Fused Anthraquinones Attenuate the Proliferation of Oral Cancer Cells through Induction of Apoptosis

To explore whether heteroarene-fused anthraquinones **2a**–**c** possess anticancer properties against oral cancer cells, we used cell impedance measurement to evaluate the cell growth profiles of two human oral squamous carcinoma cell lines with varied p53 functionality: SAS and HSC-3 cells. We chose these lines to additionally determine whether p53 functionality affects the anticancer property of these derivatives. In SAS cells, p53 is expressed as a truncated version due to an early stop codon, but its ability to be phosphorylated at serine 46 sustains its apoptotic activity (http://p53.free.fr/Database/Cancer_cell_lines/p53_cell_lines.html (accessed on 18 May 2022)). In HSC-3 cells, in contrast, a defect in the serine 46 phosphorylation of p53 in HSC-3 cells contributes to a faulty transcriptional activation of proapoptotic genes [48,49]. Our results validate that the three derivatives attenuated the growth of SAS cells, showing comparable levels of inhibition at 0.5 and 2 μM, and that the growth inhibition was similar across all three derivatives (Figure 2A–C). Contrastingly, the derivatives exhibited less cytotoxicity against p53-mutated HSC-3 cells compared to SAS cells, and **2b** displayed markedly weaker inhibition than the other two derivatives at 0.5 μM (Figure 2D–F).

Next, the mechanisms underlying the cytotoxicity of these derivatives were explored, particularly in the p53-independent scenario. This is the first report on the anticancer properties of **2a** or **2b**, while **2c** was previously shown to exert apoptotic activity on bladder cancer cells [40]. Here, we report that all three compounds provoked noticeable increases in the apoptotic populations of both cell lines at 0.5 or 2 μM, as assessed by Annexin V and PI staining (Figure 3A). The apoptotic activity of these derivatives was supported by Western blot analysis, which showed enhancement of caspase 3-directed PARP cleavage, the ADP-ribosylation of PARP, and γH_2_AX (a DNA damage marker) (Figure 3B). The uncropped blots and molecular weight markers are shown in Appendix A. The cytotoxic effect of these derivatives was also evidenced by the downregulation of anti-apoptotic Bcl-2 and c-Flip and a concurrent induction of cleaved caspase 8 in SAS p53-functional cells (Figure 3B). Interestingly, c-Flip downregulation was also observed in HSC-3 p53-mutated cells, suggesting that the studied derivatives could trigger a p53-dependent and/or p53-independent apoptotic mechanism (Figure 3B). The degradation of c-Flip is regulated by SIRT1-mediated Ku70 deacetylation, which leads to cell death [50,51]. Consistent with this, we found that the heteroarene-fused anthraquinone **2c** enhanced the acetylated Ku70 levels in both cell lines (Figure 3C). However, SIRT1 expression seemed to be unchanged in cells treated with the derivatives (Figure 3B). The lack of alteration in SIRT1 expression suggested that there may be a different (non-SIRT1) upstream regulator.

### 3.3. Bis(chloroacetamidino)heteroarene-Fused Anthraquinones Downregulate tNOX Expression at the Transcriptional and Protein Levels

To address whether the derivatives use a different upstream regulator to alter c-Flip, we explored the role of tNOX in our system, given that the tNOX–NAD^+^–SIRT1 regulatory axis contributes to the antiproliferative potency and apoptosis induction of 4,11-diaminoanthra[2,3-b]furan-5,10-diones [21,22]. Indeed, we found that tNOX expression was suppressed by the heteroarene-fused anthraquinones at the protein level in both cell lines (Figure 4A). The uncropped blots and molecular weight markers are shown in Appendix A. Moreover, pre-treatment with the proteasome inhibitor, MG132, considerably rescued the stability of tNOX, suggesting that the proteasome contributes to the tNOX degradation triggered by **2c** in both cell lines (Figure 4B). We next confirmed that tNOX was ubiquitinated in SAS cells by performing immunoprecipitation experiments with a commercially available antibody against endogenous tNOX and immunoblotting with an anti-ubiquitin antibody. It was evident that the total tNOX protein was reduced while its ubiquitin level was augmented in **2c**-exposed cells compared to controls (Figure 4C). However, the lysosome inhibitor, chloroquine, failed to reverse the 2c-triggered tNOX downregulation in HSC-3 cells, indicating that the autophagy–lysosome system might not be important for this process (Figure 4D). Derivative **2c** at 1 and 2 μM was also found to significantly reduce the transcriptional level of tNOX in SAS and HSC-3 cells (Figure 4E). These lines of evidence indicated that the heteroarene-fused anthraquinones induced tNOX transcriptional downregulation and proteasomal, but not lysosomal, degradation.

### 3.4. tNOX Acts as a Cellular Target of bis(chloroacetamidino)heteroarene-Fused Anthraquinones, as Shown by Cellular Thermal Shift Assays (CETSA) and Molecular Docking Simulations

To assess whether tNOX directly and physically binds to the studied derivatives in oral cancer cells, we conducted CETSA. This strategy is based on the concept that a ligand–protein interaction increases the thermal stability of the protein in its native configuration [52,53]. Without direct binding to its ligand, the protein is unfolded; thus, it is promptly precipitated upon heating and exhibits a lower melting temperature (*T*_m_). To calculate *T*_m_ values, we first generated CETSA curves in which the intensity of tNOX protein on Western blot was plotted against the temperature. We found that **2c** treatment increased *T*_m_ from 48.1 °C to 58.2 °C in SAS cells and from 50.3 °C to 56.2 °C in HSC-3 cells; this increase of more than 5 °C in *T*_m_ indicates that **2c** prompted the thermal stabilization of tNOX through direct binding (Figure 5A). Similarly, *T*_m_ values were increased by exposure to derivative **2a** in both cell lines, reinforcing the idea that these derivatives have tNOX-binding capacity comparable to that of **2c** (Figure 5B). The derivative **2b** was also found to target tNOX in HSC-3 cells, as evidenced by *T*_m_ increasing by the exposure to derivative **2b** (Figure 5C).

To explore the binding mode between the derivatives and the predicted tNOX structure, blind docking was performed to predict the binding sites and conduct docking calculations, which were superimposed with the different orientations (Figure 6A). Our results illustrated that the three studied derivatives might bind to the same pocket of tNOX, and enabled us to predict docking energies for **2c** (−128.1 kJ/mol), **2a** (−124.9 kJ/mol), and **2b** (−118.7 kJ/mol) (Figure 6B). Among the three compounds, **2b** seemed to have the least affinity for tNOX. The heterocyclic cores of compounds **2a**–**c** interacted via Lys 98, Pro117, Pro118, and Tyr 180 by participating in electrostatic, hydrophobic, or van der Waals interactions, while other residues were oriented in a unique manner to each derivative. We also found that the tNOX inhibitor, LY181984, exhibited a binding mode similar to those of the derivatives, although with a much lower affinity (−100.7 kJ/mol) [54] (Figure 6B). Further analysis of the binding mode revealed that the interaction residues to LY181984 were more similar to those of **2a** than those of the other two derivatives (Figure 6C). The heteroarene-fused anthraquinones were predicted to generate more hydrogen bonds than LY181984; since hydrogen bonds could help stabilize surrounding residues, this finding could explain the weaker affinity predicted for LY181984 (Figure 6).

## 4. Discussion

In this report, we describe the design and synthesis of novel antitumor heterocyclic derivatives of anthraquinone, **2a**–**c**, bearing terminal chloroacetamidines. The modification of the terminal amino groups to chloroacetamidines resulted in high potency against oral cancer cells. Importantly, we further identified tNOX as the protein target of these heteroarene-fused anthraquinones. CETSA, which has recently emerged as a powerful tool for clarifying the binding affinity of drugs to their intracellular protein targets, can be used to identify potent and selective compounds that may serve as potential therapeutic agents in an array of applications [55,56,57]. Here, we used CETSA and molecular docking simulations to identify and validate tNOX as a key protein target of these novel heteroarene-fused anthraquinones, and we propose that this targeting elicits their anticancer properties. tNOX has been implicated as a hallmark of cancer cells; it can be suppressed by anticancer drugs to provoke cell death selectively in cancer cells but not in non-transformed cells [25,32,33,58,59]. In this study, we predicted the binding model and docking energy between tNOX and the studied derivatives. Our results indicate that derivatives **2a** and **2c** have lower docking energies (and thus higher predicted affinities) toward tNOX protein, compared to **2b**. The greater binding capacities of derivatives **2a** and **2c** were supported by the results from cell impedance measurements, which were used to assess cell proliferation (Figure 2). Although derivatives **2a**–**c** exhibited similar efficacies against SAS p53-functional cells, we observed differences in their efficacies against p53-mutant HSC-3 cells. In particular, furan-fused anthraquinone **2b** had the least inhibition at 0.5 μM compared to the pyrrole-**2a** and thiophene-**2c** analogs. This suggests that the heterocyclic core critically affects the efficacy of these compounds against p53-mutated cells.

Given that tNOX functions to oxidize the reduced forms of NADH or hydroquinone, its activation increases NAD^+^ generation, and thereby accelerates proliferation, enhances migration/invasion, and alleviates apoptosis [27,28,29,30,31,32]. Compounds targeting tNOX protein have been found to suppress cancer cell survival and exhibit therapeutic efficacy [22,33,36,58,60]. However, the underlying mechanisms have not previously been assessed by protein structure predictions and analyses. Based on previous site-directed mutagenesis results [61], we postulated that the NADH-binding motif of tNOX spans from Gly590 to Leu595. The binding sites we predicted herein for the tNOX inhibitor, LY181984, are not near the envisaged NADH binding sites. LY181984 was previously shown to suppress tNOX activity at the plasma membrane of HeLa cells, and to be non-competitive or uncompetitive depending on the concentration of NADH [54]. The consistent interacting residues involved in the van der Waals interactions of the studied derivatives were predicted to be Lys98, Pro117, and Pro118, which also do not reside in the NADH binding sites (Figure 6B,C). We speculate that LY181984 uses a binding mode similar to that of the studied heteroarene-fused anthraquinones, but shows a much lower affinity (−100.7 kJ/mol) (Figure 6B). Given that hydrogen bonding can stabilize the interacted residues, and that derivatives **2a**–**c** were estimated to generate more hydrogen bonds than LY181984, we believe that the enhanced formation of hydrogen bonds may increase the binding affinity between tNOX protein and the studied derivatives [62]. Further experiments, such as efforts to co-crystallize tNOX/inhibitor, are warranted to further explore this possibility.

Another major finding of this study is our clarification that the apoptotic effect of the studied derivatives acts through c-Flip downregulation. c-Flip (cellular FADD-like interleukin 1β-converting enzyme (FLICE) inhibitory protein) acts as an anti-apoptotic protein by interfering with the activation of pro-caspase 8 in the receptor-mediated pathway [63]. Given its anti-apoptotic activity, it is not surprising that c-Flip is upregulated in many cancers and is used as a prognostic marker [64,65,66,67]. The transcriptional upregulation of c-Flip is controlled by a set of transcription factors including NFkB, CREB, and EGR1 [68,69,70]. Meanwhile, c-Flip expression is reportedly attenuated by other transcription factors such as c-Fos, FoxO3a, and c-Myc [71,72,73]. Cytoplasmically localized Ku70 (a DNA repair factor) is reported to form a complex with c-Flip to enhance its protein stability and thereby shield cells from apoptosis [51]. c-Flip has a rapid turnover rate that is mainly attributed to ubiquitin–proteasomal protein degradation [74]. Post-translational acetylation of Ku70 disrupts its interaction with c-Flip, leaving c-Flip vulnerable to ubiquitination and thereby favoring apoptosis [51,75]. The importance of Ku70 acetylation was highlighted by studies in which deficiency of the deacetylase, SIRT1, was shown to downregulate c-Flip and potentiate TRAIL-induced apoptosis by upregulating c-Myc [50,76,77]. Importantly, we demonstrated herein that the studied novel derivatives may enhance Ku70 acetylation through inhibition of the tNOX–SIRT1 axis, with concurrently increased c-Flip downregulation and provoked apoptosis in oral cancer cells, regardless of p53 functionality.

## 5. Conclusions

The three heteroarene-fused anthraquinones studied herein target and downregulate tNOX to reduce SIRT1 deacetylase activity and increase Ku70 acetylation, which triggers c-Flip degradation and induces apoptosis in oral cancer cells. Our experimental results and computational evidence suggest that heteroarene-fused anthraquinones may warrant further development as potential cancer therapeutics that act by targeting the tNOX–SIRT1 axis to provoke cancer cell death.

## Data Availability

The data presented in this study are available in this article (and Appendix A).

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
