# Peer review of "Bis(chloroacetamidino)-Derived Heteroarene-Fused Anthraquinones Bind to and Cause Proteasomal Degradation of tNOX, Leading to c-Flip Downregulation and Apoptosis in Oral Cancer Cells"

_cancers, 2022, doi:10.3390/cancers14194719_

Round 1

Reviewer 1 Report

This is a very comprehensive and high quality report of the development and use of novel compounds for cancer apoptosis.  This reviewer has no concerns regarding the approach, data or presentation, with two minor exceptions:

The simple summary is not so simple and should be rewritten with fewer or no technical terms.

The PCR primers, for example, GAAGTGTGATGCCGATAACAG, indicated in Methods should contexted, i.e., with the surrounding nucleotides, so readers can see the primer sequences in the genome browser. As it is now, these sequences are too short to “find” the relevant gene using the BLAT search tool at the genome browser.

Author Response

Thank reviewer 1 for the important suggestions. 

  1. We have edited the simple summary, it is now easier to read and understand. Please see the revised manuscript.
  2. We have provided more information regarding the primer sequences. The revision is now read as "The primers for tNOX (NCBI Reference Sequence: XM_034949457.1) were used for PCR amplifications: 5’-GCTGTGCTTCTAGGCTGTGT-3’’ (sense) and 5’- TTATCAAGACGGTGCAAGTAGGA -3’ (antisense), while GADPH (NCBI Reference Sequence: NM_001411843.1) were 5’- GGAAGGCCATGCCAGTGAGC-3’ (sense) and 5’-TATCGTGGAAGGACTCATGA-3’ (antisense). 

Reviewer 2 Report

Chang et al. report on the synthesis of 3 heterocyclic derivatives of anthraquinone an their biological activities in 2 cancer cell lines. The experimental work is solidly done and the results are apropriely analyszed. A few issues, however, will have to be addressed before publication:

Major issues:

The claim that tNOX “is universally expressed in cancer cells but not in non-transformed cells” (lines 32, 83) has to be evidenced or removed. As far as I can see refs 23-26 only look at slow-growing/resting cells and there is at least indirect literature evidence [Biochem. Pharmacol. 74 (2007) 1587– 95], that it is expressed in rapily proliferating cells like activated lymphocytes.

A list of abbreviations used should be added.

Minor issues:

Line 102: delete first “the”

Line 105: evaluate

Line 121: < 45ºC.

Line 345: provoke

Lines 348f: bis(chloroacetamidino)heteroareneanthraquinones

Author Response

Thank reviewer 2 for the valuable suggestion.

  1.  We have revised the simple summary to be easier to read and understand. In the process, we respect the reviewer's opinion and reword the description as " tNOX is a growth-related protein and is found to be expressed in cancer cells but not in non-transformed cells, and its knockdown by RNA interference in tumor cells overturns cancer phenotypes, supporting its role in cellular growth. "
  2.  A list of abbreviations is now added to the manuscript. 

    Abbreviations

    CETSA: cellular thermal shift assay

    c-Flip: Cellular FLICE-like inhibitory protein

    DIPEA: diisopropylethylamine

    ESI: electron spray 118 ionization

    HDAC: Histone deacetylase

    SIRT1: silent mating type information regulation 1 (Sirtuin 1)

    TM: melting temperature

    tNOX: Tumor-associated NADH oxidase

    Top1: topoisomerase 1

  3.  We have revised all the minor issues accordingly.